# Synthesis and In Vitro Photodynamic Activity of Cationic Boron Dipyrromethene-Based Photosensitizers Against Methicillin-Resistant *Staphylococcus aureus*

**DOI:** 10.3390/biomedicines8060140

**Published:** 2020-05-29

**Authors:** Priyanga Dharmaratne, Roy C. H. Wong, Jun Wang, Pui-Chi Lo, Baiyan Wang, Ben C. L. Chan, Kit-Man Lau, Clara B. S. Lau, Kwok-Pui Fung, Margaret Ip, Dennis K. P. Ng

**Affiliations:** 1School of Biomedical Sciences, Faculty of Medicine, The Chinese University of Hong Kong, Shatin, N.T., Hong Kong; priyanga@cuhk.edu.hk (P.D.); tinawang@cuhk.edu.hk (B.W.); kpfung@cuhk.edu.hk (K.-P.F.); 2Department of Chemistry, The Chinese University of Hong Kong, Shatin, N.T., Hong Kong; roywongchihang@gmail.com; 3Department of Biomedical Sciences, City University of Hong Kong, Tat Chee Avenue, Kowloon, Hong Kong; junma9003@hfnu.edu.cn (J.W.); gigi.lo@cityu.edu.hk (P.-C.L.); 4Institute of Chinese Medicine, The Chinese University of Hong Kong, Shatin, N.T., Hong Kong; benchan99@cuhk.edu.hk (B.C.L.C.); virginialau@cuhk.edu.hk (K.-M.L.); claralau@cuhk.edu.hk (C.B.S.L.); 5State Key Laboratory of Research on Bioactivities and Clinical Applications of Medicinal Plants, The Chinese University of Hong Kong, Shatin, N.T., Hong Kong; 6CUHK-Zhejiang University Joint Laboratory on Natural Products and Toxicology Research, The Chinese University of Hong Kong, Shatin, N.T., Hong Kong; 7Department of Microbiology, Faculty of Medicine, The Chinese University of Hong Kong, Prince of Wales Hospital, Shatin, N.T., Hong Kong; 8Shenzhen Research Institute, The Chinese University of Hong Kong, Shenzhen 518057, China

**Keywords:** antimicrobials, boron dipyrromethenes, methicillin-resistant *Staphylococcus aureus*, photodynamic therapy, photosensitizers

## Abstract

A series of cationic boron dipyrromethene (BODIPY) derivatives were synthesized and characterized with various spectroscopic methods. Having the ability to generate singlet oxygen upon irradiation, these compounds could potentially serve as photosensitizers for antimicrobial photodynamic therapy. Of the five BODIPYs being examined, the dicationic aza-BODIPY analogue (compound **5**) demonstrated the highest potency against a broad spectrum of clinically relevant methicillin-resistant *Staphylococcus aureus* (MRSA), including four ATCC-type strains (ATCC 43300, ATCC BAA-42, ATCC BAA-43, and ATCC BAA-44), two strains carrying specific antibiotic resistance mechanisms [-AAC(6’)-APH(2”) and RN4220/pUL5054], and ten non-duplicate clinical strains from hospital- and community-associated MRSAs of the important clonal types ST239, ST30, and ST59, which have previously been documented to be prevalent in Hong Kong and its neighboring countries. The *in vitro* anti-MRSA activity of compound **5** was achieved upon irradiation with near-infrared light (>610 nm) with minimal bactericidal concentrations (MBCs) ranging from 12.5 to 25 µM against the whole panel of MRSAs, except the hospital-associated MRSAs for which the MBCs were in the range of 50–100 µM. Compound **5** was significantly (*p* < 0.05) more potent than methylene blue, which is a clinically approved photosensitizer, indicating that it is a promising antimicrobial agent that is worthy of further investigation.

## 1. Introduction

*Staphylococcus aureus* remains the leading cause of hospital-associated infection and community-associated infection around the globe, where a high proportion of these infections was caused by methicillin-resistant *Staphylococcus aureus* (MRSA) [1,2]. It is the causative pathogen for most of the skin and soft tissue infections (SSTIs), endovascular infections, pneumonia, septic arthritis, endocarditis, osteomyelitis, foreign-body infections, and sepsis and is unfortunately resistant to all available penicillins and other β-lactam antibiotics [3].

In 1961, soon after the introduction of methicillin, MRSA infections were first detected in hospitals (HA-MRSA) in the United Kingdom [4]. A decline in MRSA cases was observed in the 1970s, but a dramatic re-emergence of MRSA strains occurred in the early 1980s throughout the world, including the United Kingdom, USA, Australia, and some European countries (France, Italy, and Spain) [3,4,5]. 

The scarcity of effective therapies for highly pathogenic bacterial infections, including MRSA infections, stresses the need for the development of novel approaches for either treatment or prevention of these infections. Antimicrobial photodynamic therapy (aPDT) is such an alternative approach that is currently being explored as a potential therapeutic treatment option for various types of infections, including bacterial, fungal, viral, or even parasitic in nature [6,7,8,9,10]. It is conceptually simple and only requires a photosensitizer to generate reactive oxygen species (ROS), such as singlet oxygen, upon illumination with visible or near-infrared light [11]. This new treatment modality offers various advantages over the existing antimicrobial therapies. First, aPDT can combat multidrug-resistant strains due to its unique cytotoxic mechanism [12,13]. Second, it does not induce cytotoxicity in the absence of light irradiation. Hence, toxicity is largely confined to the photosensitizer-located and light-irradiated zone [14]. In addition, the inactivation of microorganisms by this method is usually rapid [15] and the photosensitizers show limited or no resistance so far [16,17,18].

While various classes of photosensitizers, including the tetrapyrrole-based macrocycles (e.g., porphyrins, bacteriochlorins, and phthalocyanines) [19,20,21], phenothiazine derivatives (e.g., methylene blue and Toludine Blue O) [22,23], and natural product derivatives (e.g., curcumin, riboflavin, and hypericin) [24,25,26], have been studied for their anti-MRSA activities, relatively little is known regarding the application of boron dipyrromethene (BODIPY)-based photosensitizers against MRSA infections. This class of functional dyes possesses tunable absorption and photophysical properties, ease of chemical modification, and high stability, and these dyes are therefore promising photosensitizers for PDT [27,28,29,30]. Frimannsson et al. reported a brominated aza-BODIPY and its non-brominated analogue for aPDT [31]. The former showed 6.8 and 3.4 log_10_ colony-forming unit (CFU) reduction against methicillin-sensitive *Staphylococcus aureus* (MSSA) and MRSA strains, respectively, at a drug dose of 5 µg/mL with a light fluence of 16 J/cm^2^. By contrast, the non-brominated counterpart was totally ineffective at the same concentration even when the fluence was increased to 25 J/cm^2^. Carpenter et al. reported another aza-BODIPY derivative that was photodynamically active toward a broad spectrum of microbials including viruses, fungi, and bacteria [32]. It exhibited 5–6 log_10_ CFU reduction against MRSA ATCC-44 at a concentration as low as 0.1 µM. However, relatively few clinically relevant MRSAs have been explored, and there is a lack of understanding of this new class of photosensitizers against MRSA infections. We report herein the synthesis and photophysical properties of a series of cationic BODIPY derivatives and their *in vitro* antibacterial photodynamic activity against a broad spectrum of clinically relevant MRSA.

## 2. Materials and Methods

### 2.1. General Information

All the reactions were performed under an atmosphere of nitrogen. Tetrahydrofuran (THF), CH_2_Cl_2_, and *N*,*N*-dimethylformamide (DMF) were purified with an INERT solvent purification system (PS-MD-5, Amesbury, MA, USA), while all other solvents were used without further purification. Silica gel (Macherey-Nagel, Düren, Germany; 230–400 mesh) was used as the stationary phase for chromatographic purification. Compounds **6** [33], **7** [34], **13** [35], **15** [36], **16** [37], and **17** [38] were prepared according to the literature procedures.

^1^H and ^13^C{^1^H} NMR spectra were recorded on a Bruker AVANCE III 500 spectrometer (^1^H, 500 MHz; ^13^C, 125.0 MHz) or a Bruker AVANCE III 400 spectrometer (Billerica, MA, USA) (^1^H, 400 MHz; ^13^C, 100.6 MHz) in CDCl_3_ or DMSO-*d*_6_. Spectra were referenced internally by using the residual solvent (^1^H, δ = 7.26 (for CDCl_3_), δ = 2.50 (for DMSO-*d*_6_)) or solvent (^13^C, δ = 77.2 (for CDCl_3_), δ = 39.5 (for DMSO-*d*_6_)) resonances relative to SiMe_4_. Electrospray ionization (ESI) and matrix-assisted laser desorption/ionization time-of-flight (MALDI-TOF) mass spectra were recorded on a Thermo QEF MS mass spectrometer (Waltham, MA, USA) and a Bruker Autoflex speed MALDI-TOF mass spectrometer (Billerica, MA, USA), respectively. Electronic absorption and steady-state fluorescence spectra were taken on a Shimazu UV-1800 UV-Vis spectrophotometer (Tokyo, Tokyo Prefecture, Japan) and a Horiba FluoroMax spectrofluorometer (Kyoto, Kyoto Prefecture, Japan), respectively. 

### 2.2. Synthesis of BODIPY Derivatives

Preparation of compound **8**. Knoevenagel condensation of compound **6** (50 mg, 95 µmol) and compound **7** (56 mg, 0.21 mmol) in the presence of glacial acetic acid (0.2 mL, 4.6 mmol), piperidine (0.25 mL, 2.9 mmol), and a small amount of Mg(ClO_4_)_2_ was carried out in toluene (50 mL). The mixture was heated under reflux for 16 h, during which the water formed was removed azeotropically with a Dean–Stark apparatus. After evaporation in vacuo, the residue was purified by column chromatography using CH_2_Cl_2_/MeOH (20:1, *v*/*v*) as the eluent (35 mg, 36%). ^1^H NMR (400 MHz, CDCl_3_) δ 8.08 (d, *J* = 16.8 Hz, 2 H, C=CH), 7.59–7.63 (m, 6 H, ArH and C=CH), 7.04 (d, *J* = 8.4 Hz, 2 H, ArH), 6.96 (d, *J* = 8.8 Hz, 4 H, ArH), 6.79 (d, *J* = 8.0 Hz, 2 H, ArH), 4.19 (t, *J* = 4.8 Hz, 4 H, CH_2_), 3.89 (t, *J* = 4.8 Hz, 4 H, CH_2_), 6.75–3.77 (m, 4 H, CH_2_), 3.66–3.71 (m, 8 H, CH_2_), 3.55–3.57 (m, 4 H, CH_2_), 3.39 (s, 6 H, CH_3_), 3.04 (s, 6 H, NCH_3_), 1.52 (s, 6 H, CH_3_). ^13^C{^1^H} NMR (CDCl_3_) δ 14.2, 40.5, 59.2, 67.7, 69.8, 70.7, 70.8, 71.0, 72.1, 109.9, 112.6, 115.1, 116.4, 122.0, 129.3, 129.4, 130.1, 132.8, 138.4, 140.6, 141.3, 148.0, 151.0, 160.0. HRMS (ESI) Calcd. for C_49_H_58_BBr_2_F_2_N_3_NaO_8_ [M+Na]^+^ 1048.2538, found 1048.2544.

Preparation of compound **1**. A mixture of compound **8** (15 mg, 15 µmol) and CH_3_I (104 mg, 0.73 mmol) in DMF (10 mL) was stirred at room temperature overnight. The product was obtained after precipitation in hexane followed by filtration and dried in vacuo (14 mg, 82%). ^1^H NMR (500 MHz, DMSO-*d*_6_) δ 8.21 (d, *J* = 8.5 Hz, 2 H, ArH), 8.10 (d, *J* = 16.5 Hz, 2 H, C=CH), 7.84 (d, *J* = 8.5 Hz, 2 H, ArH), 7.63 (d, *J* = 8.5 Hz, 4 H, ArH), 7.49 (d, *J* = 16.5 Hz, 2 H, C=CH), 7.10 (d, *J* = 8.5 Hz, 4 H, ArH), 4.19 (t, *J* = 4.5 Hz, 4 H, CH2), 3.78 (t, *J* = 4.5 Hz, 4 H, CH_2_), 3.70 (s, 9 H, NCH_3_), 3.60–3.62 (m, 4 H, CH_2_), 3.53-3.56 (m, 8 H, CH_2_), 3.43–3.45 (m, 4 H, CH_2_), 3.25 (s, 6 H, CH_3_), 1.40 (s, 6 H, CH_3_). ^13^C{^1^H} NMR (DMSO-*d*_6_) δ 13.8, 56.6, 58.1, 67.5, 68.9, 69.7, 69.9, 70.0, 71.3, 109.9, 115.0, 115.4, 121.8, 128.7, 129.2, 130.5, 131.4, 135.7, 137.3, 139.0, 140.8, 147.9, 148.4, 160.2. HRMS (ESI) Calcd. for C_50_H_61_BBr_2_F_2_N_3_NaO_8_ [M+Na]^2+^ 531.6384, found 531.6384.

Preparation of compound **10**. A mixture of compound **6** (50 mg, 95 µmol), compound **9** (26 mg, 0.21 mmol), glacial acetic acid (0.2 mL, 4.6 mmol), piperidine (0.25 mL, 2.9 mmol), and a small amount of Mg(ClO_4_)_2_ in toluene (50 mL) was refluxed for 16 h. The water formed during the reaction was removed azeotropically with a Dean–Stark apparatus. The mixture was concentrated under reduced pressure. The residue was purified by column chromatography using CH_2_Cl_2_/MeOH (10:1, *v*/*v*) as the eluent (29 mg, 42%). ^1^H NMR (400 MHz, DMSO-*d*_6_) δ 10.07 (s, 2 H, OH), 8.00 (d, *J* = 16.8 Hz, 2 H, C=CH), 7.49 (d, *J* = 8.4 Hz, 4 H, ArH), 7.39 (d, *J* = 16.8 Hz, 2 H, C=CH), 7.15 (d, *J* = 8.4 Hz, 2 H, ArH), 6.88 (d, *J* = 8.4 Hz, 4 H, ArH), 6.85 (d, *J* = 8.8 Hz, 2 H, ArH), 3.00 (s, 6 H, NCH_3_), 1.48 (s, 6 H, CH_3_). ^13^C{^1^H} NMR (DMSO-*d*_6_) δ 13.8, 54.9, 109.2, 112.1, 114.2, 116.2, 120.3, 127.2, 129.2, 132.1, 138.6, 140.8, 140.9, 147.1, 150.9, 159.4, 162.3. HRMS (ESI) Calcd. for C_35_H_29_BBr_2_F_2_N_3_O_2_ [M-H]^−^ 732.0684, found 732.0682.

Preparation of compound **12**. A mixture of compound **10** (20 mg, 27 µmol), compound **11** (10 mg, 0.12 mmol), and a catalytic amount of *p*-toluenesulfonic acid (TsOH) in CH_2_Cl_2_ (10 mL) was stirred for 2 h. Et_3_N was added to neutralize the acid and the volatile was removed under reduced pressure. The product was obtained after purification by column chromatography using CH_2_Cl_2_/MeOH (50:1, *v*/*v*) as the eluent (24 mg, 98%). ^1^H NMR (400 MHz, CDCl_3_) δ 8.08 (d, *J* = 16.8 Hz, 2 H, C=CH), 7.60–7.65 (m, 6 H, ArH and C=CH), 7.10 (d, *J* = 8.8 Hz, 4 H, ArH), 7.05 (d, *J* = 8.4 Hz, 2 H, ArH), 6.80 (d, *J* = 8.4 Hz, 2 H, ArH), 5.50 (t, *J* = 2.8 Hz, 2 H, CH), 3.88–3.94 (m, 2 H, CH), 3.61–3.66 (m, 2 H, CH), 3.05 (s, 6 H, NCH_3_), 1.99–2.07 (m, 2 H, CH), 1.88–1.91 (m, 4 H, CH_2_), 1.63–1.72 (m, 6 H, CH_2_), 1.53 (s, 6 H, CH_3_). ^13^C{^1^H} NMR (CDCl_3_) δ 14.2, 18.8, 25.3, 30.4, 40.5, 62.2, 96.3, 110.0, 112.6, 116.7, 116.9, 129.2, 129.4, 130.8, 132.9, 138.5, 140.7, 141.3, 148.1, 151.0, 158.2. HRMS (ESI) Calcd. for C_45_H_47_BBr_2_F_2_N_3_O_4_ [M+H]^+^ 902.1981, found 902.1970.

Preparation of compound **2**. A mixture of compound **12** (15 mg, 17 µmol) and CH_3_I (118 mg, 0.83 mmol) in DMF (10 mL) was stirred at room temperature overnight. The product was obtained after precipitation in hexane followed by filtration and dried in vacuo (13 mg, 90%). ^1^H NMR (500 MHz, DMSO-*d*_6_) δ 10.14 (s, 2 H, OH), 8.19 (d, *J* = 9.0 Hz, 2 H, ArH), 8.04 (d, *J* = 16.5 Hz, 2 H, C=CH), 7.79 (d, *J* = 9.0 Hz, 2 H, ArH), 7.51 (d, *J* = 9.0 Hz, 4 H, ArH), 7.40 (d, *J* = 16.5 Hz, 2 H, C=CH), 6.89 (d, *J* = 9.0 Hz, 4 H, ArH), 3.70 (s, 9 H, NCH_3_), 1.32 (s, 6 H, CH_3_). ^13^C{^1^H} NMR (DMSO-*d*_6_) δ 13.7, 56.7, 109.8, 114.0, 116.4, 121.8, 127.2, 129.5, 130.5, 131.3, 135.8, 136.7, 139.4, 140.5, 147.9, 148.4, 159.7. HRMS (ESI) Calcd. for C_36_H_33_BBr_2_F_2_N_3_O_2_ [M]^+^ 748.0984, found 748.0973.

Preparation of compound **14**. A mixture of compound **6** (50 mg, 95 µmol), compound **13** (50 mg, 0.21 mmol), glacial acetic acid (0.2 mL, 4.6 mmol), piperidine (0.25 mL, 2.9 mmol), and a small amount of Mg(ClO_4_)_2_ in toluene (50 mL) was refluxed for 16 h. The water formed during the reaction was removed azeotropically with a Dean–Stark apparatus. The mixture was concentrated under reduced pressure. The residue was purified by column chromatography using CH_2_Cl_2_/MeOH (20:1, *v*/*v*) as the eluent (33 mg, 36%). ^1^H NMR (500 MHz, CDCl_3_) δ 8.08 (d, *J* = 16.5 Hz, 2 H, C=CH), 7.60–7.63 (m, 6 H, ArH and C=CH), 7.07 (d, *J* = 8.5 Hz, 2 H, ArH), 6.96 (d, *J* = 8.5 Hz, 4 H, ArH), 6.84 (d, *J* = 7.5 Hz, 2 H, ArH), 4.51 (quintet, *J* = 6.0 Hz, 2 H, CH), 4.20 (dd, *J* = 6.5 and 8.5 Hz, 2 H, CH), 4.12 (dd, *J* = 5.5 and 9.5 Hz, 2 H, CH), 4.01 (dd, *J* = 5.5 and 9.5 Hz, 2 H, CH), 3.93 (dd, *J* = 6.0 and 8.5 Hz, 2 H, CH), 3.06 (s, 6 H, NCH_3_), 1.53 (s, 6 H, CH_3_), 1.49 (s, 6 H, CH_3_), 1.42 (s, 6 H, CH_3_). ^13^C{^1^H} NMR (CDCl_3_) δ 14.2, 25.5, 27.0, 40.7, 43.8, 67.0, 69.0, 74.1, 110.0, 112.9, 115.0, 116.6, 129.4, 130.4, 132.9, 138.4, 140.6, 141.4, 148.1, 159.7, 159.9 (some of the signals were overlapped). HRMS (ESI) Calcd. for C_47_H_49_BBr_2_F_2_N_3_O_6_ [M-H]^−^ 960.2054, found 960.2047.

Preparation of compound **3**. A mixture of compound **14** (15 mg, 16 µmol) and CH_3_I (111 mg, 0.78 mmol) in DMF (10 mL) was stirred at room temperature overnight. The product was obtained after precipitation in hexane followed by filtration and dried in vacuo (13 mg, 80%). ^1^H NMR (400 MHz, DMSO-*d*_6_) δ 8.22 (d, *J* = 9.2 Hz, 2 H, ArH), 8.09 (d, *J* = 16.8 Hz, 2 H, C=CH), 7.83 (d, *J* = 8.8 Hz, 2 H, ArH), 7.62 (d, *J* = 8.8 Hz, 4 H, ArH), 7.48 (d, *J* = 16.8 Hz, 2 H, C=CH), 7.09 (d, *J* = 8.8 Hz, 4 H, ArH), 5.02–5.03 (m, 1 H), 4.71–4.74 (m, 1 H), 4.09 (dd, *J* = 4.0 and 10.0 Hz, 2 H, CH), 3.95 (dd, *J* = 6.0 and 10.0 Hz, 2 H, CH), 3.81–3.84 (m, 2 H, CH), 3.71 (s, 9 H, NCH_3_), 3.47 (d, *J* = 6.0 Hz, 4 H, CH_2_), 1.37 (s, 6 H, CH_3_). HRMS (ESI) Calcd. for C_42_H_45_BBr_2_F_2_N_3_O_6_ [M]^+^ 896.1722, found 896.1720.

Preparation of compound **4**. According to the procedure described for compound **8**, compound **15** (0.25 g, 0.34 mmol) was treated with compound **16** (0.18 g, 0.75 mmol), glacial acetic acid (0.40 mL, 9.2 mmol), piperidine (0.50 mL, 5.8 mmol), and a small amount of Mg(ClO_4_)_2_ in toluene (50 mL) under reflux for 3 h. CH_2_Cl_2_/MeOH (50:1, *v*/*v*) was used as the eluent for purification by column chromatography. The green fraction was collected and concentrated under reduced pressure. Without further purification, the crude intermediate product was further treated with CH_3_I in THF at room temperature overnight. The product was obtained after precipitation in hexane followed by filtration and dried in vacuo (187 mg, 38%). ^1^H NMR (400 MHz, DMSO-*d*_6_) δ 8.07 (d, *J* = 16.4 Hz, 2 H, C=CH), 7.62 (d, *J* = 8.8 Hz, 4 H, ArH), 7.43 (d, *J* = 16.4 Hz, 2 H, C=CH), 7.34 (d, *J* = 8.8 Hz, 2 H, ArH), 7.17 (d, *J* = 8.8 Hz, 2 H, ArH), 7.09 (d, *J* = 8.8 Hz, 4 H, ArH), 4.15–4.19 (m, 6 H, CH_2_), 3.81 (br s, 2 H, CH_2_), 3.51–3.64 (m, 10 H, CH_2_), 3.40–3.46 (m, 10 H, CH_2_), 3.25 (s, 3 H, CH_3_), 2.98 (s, 6 H, CH_3_), 2.17 (br s, 4 H, CH_2_), 1.48 (s, 6 H, CH_3_), 1.25 (t, *J* = 7.2 Hz, 12 H, CH_3_).

Preparation of compound **18**. A mixture of compound **17** (200 mg, 0.23 mmol), 1,6-dibromohexane (228 mg, 0.93 mmol), and K_2_CO_3_ (129 mg, 0.93 mmol) in DMF (20 mL) was stirred at 70–80 °C for 8 h. After cooling, the volatile was evaporated in vacuo. The residue was re-dissolved in CH_2_Cl_2_ (50 mL), washed with water (50 mL × 3), and concentrated under reduced pressure. The residue was purified by column chromatography using CHCl_3_/ethyl acetate (2:1 *v*/*v*) as the eluent to give the product as a dark blue solid (179 mg, 65%). ^1^H NMR (400 MHz, CDCl_3_) δ 8.08-8.05 (m, 8 H, ArH), 6.99–7.04 (m, 8 H, ArH), 6.95 (s, 2 H, pyrrole-H), 4.26 (t, *J* = 4.8 Hz, 4 H, OCH_2_), 4.07 (t, *J* = 6.4 Hz, 4 H, OCH_2_), 3.95 (t, *J* = 4.8 Hz, 4 H, OCH_2_), 3.79–3.82 (m, 4 H, OCH_2_), 3.73–3.76 (m, 4 H, OCH_2_), 3.69–3.71 (m, 4 H, OCH_2_), 3.58–3.60 (m, 4 H, OCH_2_), 3.47 (t, *J* = 6.8 Hz, 4 H, CH_2_Br), 3.41 (s, 6 H, OCH_3_), 1.92–1.96 (m, 4 H, CH_2_), 1.84–1.87 (m, 4 H, CH_2_), 1.55 (br s, 8 H, CH_2_).

Preparation of compound **19**. A mixture of compound **18** (160 mg, 0.13 mmol), 4,4’-dipyridine (847 mg, 5.42 mmol), NaI (203 mg, 1.35 mmol), and NaHCO_3_ (187 mg, 2.23 mmol) in acetone (30 mL) was stirred under reflux for 3 days. After cooling, the solvent was removed under reduced pressure. The residue was re-dissolved in CHCl_3_ (50 mL), washed with water (50 mL × 3), and concentrated under reduced pressure. The mono-substituted aza-BODIPY was removed by column chromatography using CH_2_Cl_2_/MeOH (10:1 *v*/*v*) as the eluent. The desired product (90 mg, 42%) was obtained by changing the eluent to CH_3_NO_2_/2M aq. NH_4_Cl/MeOH/acetone (5:10:50:35 *v*/*v*/*v*/*v*). ^1^H NMR (400 MHz, CDCl_3_) δ 9.58 (br s, 4 H, ArH), 8.76 (br s, 4 H, ArH), 8.28 (br s, 4 H, ArH), 8.05 (d, *J* = 8.0 Hz, 4 H, ArH), 7.96 (d, *J* = 8.0 Hz, 4 H, ArH), 7.66 (br s, 4 H, ArH), 7.02 (d, *J* = 8.0 Hz, 4 H, ArH), 6.91 (br s, 2 H, pyrrole-H), 6.82 (d, *J* = 8.0 Hz, 4 H, ArH), 5.02 (br s, 4 H, CH_2_N^+^), 4.24 (br s, 4 H, OCH_2_), 3.94 (br s, 8 H, OCH_2_), 3.80 (br s, 4 H, OCH_2_), 3.74 (br s, 4 H, OCH_2_), 3.69 (br s, 4 H, OCH_2_), 3.59 (br s, 4 H, OCH_2_), 3.41 (s, 6 H, OCH_3_), 2.06 (br s, 4 H, CH_2_), 1.75 (br s, 4 H, CH_2_), 1.43–1.52 (m, 8 H, CH_2_). MS (ESI): 666.2 (100%, M^2+^).

Preparation of compound **5**. A mixture of compound **19** (30 mg, 0.02 mmol) and bromine (20 μL, 0.39 mmol) in CH_2_Cl_2_ (10 mL) was stirred at room temperature for 40 min. The reaction was monitored by UV-Vis spectrophotometer as the absorption band of compound **19** was blue-shifted after the addition of bromine atoms at the pyrrole rings. The solution was then diluted with CH_2_Cl_2_ and washed with Na_2_S_2_O_4_ aqueous solution and water twice. The crude product was purified by recrystallization from CH_2_Cl_2_ and hexane as a dark blue solid (20 mg, 65%). ^1^H NMR (400 MHz, CDCl_3_) δ 9.07 (br s, 4 H, ArH), 8.66 (br s, 4 H, ArH), 8.26 (br s, 4 H, ArH), 7.81 (br s, 4 H, ArH), 7.66 (br s, 4 H, ArH), 7.58 (br s, 4 H, ArH), 6.91 (br s, 4 H, ArH), 6.66 (br s, 4 H, ArH), 4.67 (br s, 4 H, CH_2_N^+^), 4.16 (br s, 4 H, OCH_2_), 3.88 (br s, 4 H, OCH_2_), 3.74–3.76 (m, 4 H, OCH_2_), 3.68–3.70 (m, 4 H, OCH_2_), 3.64–3.66 (m, 8 H, OCH_2_), 3.53–3.55 (m, 4 H, OCH_2_), 3.36 (s, 6 H, OCH_3_), 1.96 (br s, 4 H, CH_2_), 1.58 (br s, 4 H, CH_2_), 1.34 (br s, 8 H, CH_2_). ^13^C{^1^H} NMR (CDCl_3_) δ 15.3, 25.5, 25.8, 28.8, 31.5, 59.2, 61.5, 66.8, 67.6, 67.9, 69.7, 69.9, 70.7, 70.8, 71.0, 72.0, 108.8, 113.8, 114.3, 122.1, 123.5, 126.0, 132.5, 141.5, 142.2, 144.1, 145.7, 150.9, 152.7, 157.1, 160.3, 160.9 (some of the signals were overlapped). HRMS (ESI) Calcd. for C_78_H_86_BBr_2_F_2_N_7_O_10_ [M]^2+^ 744.7428, found 744.7453.

### 2.3. Determination of Fluorescence Quantum Yields

The fluorescence quantum yields were determined according to the previously described method [39] using unsubstituted zinc(II) phthalocyanine (ZnPc) in DMF as the reference (Φ_F_ = 0.28).

### 2.4. Determination of Singlet Oxygen Quantum Yields

To determine the singlet oxygen quantum yields of the BODIPY derivatives, they (at 1 µM) were mixed with the singlet oxygen probe 1,3-diphenylisobenzofuran (DPBF) (30 µM) in DMF. The decrease in absorbance of DPBF at 415 nm was monitored during irradiation. The light source consisted of a 100 W halogen lamp, a water tank for cooling, and a color filter with a cut-on wavelength at 610 nm (Newport, Irvine, CA, USA). The singlet oxygen quantum yields (Φ_Δ_) of the samples were determined by the equation Φ_Δ_(sample) = (W_sample_/W_ref_)(I_ref_/I_sample_)Φ_Δ_(ref), where W and I are the DPBF photobleaching rate and the rate of light absorption, respectively [40]. Unsubstituted ZnPc in DMF was used as the reference (Φ_Δ_ = 0.56).

### 2.5. Bacterial Strains and Culture Conditions

The bacterial strains, including four ATCC-type MRSA (ATCC 43300, ATCC BAA-42, ATCC BAA-43, and ATCC BAA-44) and two antibiotic resistant SA [AAC(6’)-APH(2”) and RN4220/pUL5054] strains were used for the study. The AAC(6’)-APH(2”)-strain expresses the bi-functional enzyme AAC(6’)-APH(2”), which is an aminoglycoside-modifying enzyme conferring high-level gentamicin resistance (minimum inhibitory concentration (MIC): >128 μg/mL). The RN4220/pUL5054 strain over-expresses the *msr*(A) gene encoding for an ATP-binding cassette (ABC) transporter that induces resistance against erythromycin (MIC: 128 μg/mL) [41]. Ten non-duplicate clinical isolates, namely 5 hospital-associated (HA) and 5 community-associated (CA) MRSAs, were included. They include the important clonal types ST239, ST30, and ST59 previously documented to be prevalent in Hong Kong and in neighboring countries [42,43,44]. All MRSA strains were grown in Mueller–Hinton Broth (MHB) for 18 h at 37 °C. The overnight culture suspension was adjusted to McFarland 0.5 and suspended in MHB to make a final concentration of 1.0 × 10^6^ CFU/mL. Altogether, 16 MRSA strains (6 ATCC-type strains and 10 clinical non-duplicate isolates) were included for *in vitro* aPDT studies.

### 2.6. In Vitro Photodynamic Minimal Bactericidal Concentration (PD-MBC) Studies

Minimal bactericidal concentrations (MBCs) of BODIPY derivatives **1**–**5** and methylene blue were determined according to the Clinical and Laboratory Standards Institute (CLSI) guidelines [45] against 16 strains in 96-well microtiter plates. Briefly, photosensitizer solutions for PDT study were prepared freshly by dissolving compounds **1**–**5** in dimethyl sulfoxide (DMSO) to make 10 mM stock solutions. They were then diluted with Tween 80 and MHB by a serial two-fold dilution procedure to obtain final working concentrations. Tween 80 and DMSO concentrations were maintained at or below 0.1% and 1% (*v*/*v*), respectively. Aliquots of this suspension (200 μL) was incubated at 37 °C for 120 min in the dark. Selected plates were illuminated from above with light intensity 40 mW/cm^2^ using a 300 W quartz-halogen lamp attenuated by a 5 cm layer of water (heat buffer) and a color glass filter with a cut-on wavelength of 610 nm (65CGA-610, Newport, Franklin, MA, USA). The intrinsic toxicity (dark toxicity) of each compound was determined by samples of each microbial suspension that were incubated with each compound separately in the dark for 140 min, corresponding to the pre-irradiation and illumination times. The effect of PDT alone was verified by adding 100 μL of PBS and irradiating for 20 min (48 J/cm^2^). The untreated control group (negative control) did not receive any photosensitizers nor light. Solvent toxicity (blank control) was also evaluated with 0.1% Tween 80 and 1% DMSO (*v*/*v*) to mimic the *in vitro* aPDT assay. The positive control groups were incubated with varied concentrations of methylene blue for 120 min followed by light illumination for 20 min. After the PDT, the plates were re-incubated at 37 °C overnight under dark conditions. To determine the MBC, the treated broth culture from wells that did not show any visible growth was cultured (10 µL) on freshly prepared sterile blood agar plates. The least concentration (highest dilution) of the compound that completely inhibited colony formation on a solid agar medium after incubation at 35 °C for 24 h was considered as MBC. Each experiment was carried out in triplicate and the range of MBC values was reported.

## 3. Results and Discussion

### 3.1. Synthesis of Cationic BODIPY Derivatives

A series of mono- and di-cationic BODIPY derivatives (compounds **1**–**5**) were designed and synthesized. Other than the cationic moieties, different hydrophilic groups were also introduced to increase the water solubility of the compounds and allow study of their effects on the aPDT efficiency. Scheme 1 shows the synthetic routes used to prepare distyryl BODIPYs **1**–**3** using the previously reported BODIPY **6** as a starting material. Knoevenagel condensation of compound **6** with the triethylglycol-substituted benzaldehyde **7** led to the formation of the distyryl analogue **8**, which was then *N*-methylated with CH_3_I in DMF to afford the ammonium distyryl BODIPY **1**. Similarly, by using 4-hydroxybenzaldehyde (**9**) to condense with compound **6** using a molecular sieve as the dehydrating agent, the dihydroxy distyryl BODIPY **10** was obtained. The hydroxyl groups of this compound were then protected by treating the compound with 3,4-dihydro-2H-pyran (**11**) in the presence of a catalytic amount of TsOH. The resulting compound **12** was then treated with CH_3_I in DMF to give compound **2**, in which the protective groups were removed under this reaction condition. To study the effect of hydroxy groups, the tetrahydroxy analogue **3** was also prepared. Knoevenagel condensation of compound **6** with the isopropylidene glycerol-substituted benzaldehyde **13** gave distyryl BODIPY **14**. Upon treatment with CH_3_I in DMF, the compound underwent one-pot deprotection and *N*-methylation to afford the target compound **3**.

To study the effect of cationic moieties on the aPDT efficiency, we also prepared two dicationic analogues (compounds **4** and **5**). As shown in Scheme 2, treatment of the previously reported BODIPY **15** with 4-[3-(diethylamino)propoxy]benzaldehyde (**16**) under typical reaction conditions of Knoevenagel condensation resulted in the formation of the distyryl BODIPY intermediate product, which was not purified and *N*-methylated directly in THF to give the dicationic analogue **4**. Apart from distyryl BODIPYs, we also studied the aza-BODIPY derivatives that are also promising near-infrared-absorbing photosensitizers [46]. The synthetic route used to prepare the aza-BODIPY analogue **5** is shown in Scheme 3. *O*-Alkylation of compound **17** with an excess of 1,6-dibromohexane in the presence of K_2_CO_3_ in DMF led to the formation of compound **18**, which underwent further nucleophilic substitution with 4,4′-dipyridine in the presence of NaI and NaHCO_3_ in acetone to afford the dicationic analogue **19**. Finally, bromination of compound **19** using bromine in CH_2_Cl_2_ gave the expected product **5**. The introduction of these heavy bromine atoms is essential in order to enable the compound to generate singlet oxygen by promoting the intersystem crossing through the heavy-atom effect.

### 3.2. Spectroscopic and Photophysical Properties

All the new compounds were characterized with various spectroscopic methods, including ^1^H and ^13^C NMR spectroscopy (Appendix A-S11), as well as high-resolution mass spectrometry. The distyryl BODIPYs **1**–**4** and aza-BODIPY **5** showed a strong absorption in the near-infrared region (661–680 nm) in DMF, which is a desirable feature of photosensitizers that can prevent absorption by biomolecules and facilitate light penetration into tissues. Upon excitation at 610 nm, these compounds showed a fluorescence emission at 679–719 nm. For the distyryl BODIPY series, the diiodo analogue **4** showed a slightly blue-shifted absorption band and significantly weaker fluorescence than the dibromo counterparts **1**–**3**, which could be attributed to the heavy-atom effect. As summarized in Table 1, the absorption band of BODIPY **4** was blue-shifted to 661 nm and its fluorescence quantum yield (Φ_F_) was only 0.21 with reference to ZnPc in DMF (Φ_F_ = 0.28), whereas the brominated analogues showed a red-shifted absorption band (up to 680 nm for BODIPY **2**) and higher Φ_F_ value (up to 0.42 for BODIPY **3**). The spectral features of the aza-BODIPY **5** were remarkably different. The Stokes shift (14 nm vs. up to 39 nm for **2**) and fluorescence quantum yield (0.01 vs. up to 0.42 for BODIPY **3**) were significantly lower than those of BODIPYs **1**–**4**. To evaluate their singlet oxygen generation efficiency, DPBF was used as the singlet oxygen scavenger. By monitoring the rate of photodegradation of this quencher in DMF, as reflected by the decrease in its absorbance at 415 nm, the singlet oxygen quantum yield (Φ_Δ_) could be determined. As shown in Table 1, the weakly fluorescent distyryl BODIPY **4** and aza-BODIPY **5** showed significantly higher singlet oxygen quantum yield than the brominated analogues **1**–**3** as a result of the heavy-atom effect due to the iodo substituents. 

### 3.3. Assessment of In Vitro PD-MBC

The photoinduced antibacterial activities of these BODIPY derivatives (compounds **1**–**5**) were then examined against 16 MRSA strains by measuring the photodynamic minimal bactericidal concentrations (PD-MBC). The highest concentration tested for each photosensitizer was set as 100 µM, since it was believed that higher concentrations would not be feasible for further in vivo investigations. The pre-incubation time and illumination time were fixed at 120 and 20 min, respectively, for the aPDT studies as determined by our previous investigations. It was found that the mono-cationic distyryl BODIPYs **1**–**3** showed comparatively lower (*p* < 0.05) anti-MRSA activity (Table 2), which could be a result of their lower singlet oxygen quantum yields (0.07–0.09, Table 1). Surprisingly, despite the fact that the dicationic analogue **4** was an efficient singlet oxygen generator in DMF (Φ_Δ_ = 0.54, Table 1), its potency against the MRSA strains was also low (Table 2). By contrast, the aza-BODIPY **5** showed significantly lower (*p* < 0.05) MBC values against all the tested MRSA strains (except hospital-associated (HA) W231 strain), regardless of their respective antibiotic resistance and microbial type (Table 2). At concentrations of ≤100 μM, it could completely eradicate (PD-MBC) all the 16 MRSAs after illumination for 20 min (λ > 610 nm, ~40 mW/cm^2^). For comparison, the benchmark photosensitizer methylene blue showed no statistically significant cell eradication under these conditions, and could only eradicate some of the strains when greater than 6–100-fold higher concentrations were used (typically 625 μM or higher, Table 2).

## 4. Conclusions

A series of cationic BODIPY derivatives were synthesized and evaluated for their antimicrobial photodynamic activity against a broad spectrum of MRSAs. They exhibited a strong absorption in the near-infrared region (661–680 nm) and could generate singlet oxygen in DMF. The singlet oxygen quantum yields of the diiodo analogues **4** and **5** were particularly high (0.54 and 0.42, respectively) owing to the heavy-atom effect. Upon irradiation, the aza-BODIPY analogue **5** showed high potency against all the 16 MRSA strains examined, including several antibiotic-resistant and HA and CA strains, with MBCs ranging from 12.5 to 25 µM (or 50–100 µM for the HA strains). Its aPDT efficiency was significantly higher than that of the clinically approved photosensitizer MB. The results suggested that aza-BODIPY **5** is a potential antimicrobial photodynamic therapeutic agent that is worthy of further investigation.

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
