# Peer review of "Synthesis and In Vitro Photodynamic Activity of Cationic Boron Dipyrromethene-Based Photosensitizers Against Methicillin-Resistant Staphylococcus aureus"

_biomedicines, 2020, doi:10.3390/biomedicines8060140_

Round 1
Reviewer 1 Report
The paper is interesting from both experimental and practical point of view and it could be potentialy published in Biomedicines, but it needs some minor revisions before final acceptation.
In Abstract in line 8 the statement "and 5 each from hospital" is not clear and it should be explained or corrected, while in line 11 the statement "activity of compound 5" should be used instead of "activity of 5".
Chapter 3 Materials and Methods should be placed between chapter 1 Introduction and chapter 2 Results and Discussion.
As table 1 should be self-explanatory all abbreviations should be explained.
In table 2 in the title and in description of particular columns the statements compound 1, compound 2, compound 3, compound 4 and compound 5 should be used instead of 1-5.
In subchapter 3.2 Synthesis of BODIPY derivatives the statements "Preparation of compound 8 (and similarly 1, 10, 12, 2, 14, 3, 4, 18, 19, 5)" should be used instead of "Preparation of 8 (and similarly 1, 10, 12, 2, 14, 3, 4, 18, 19, 5)1, 10, 12, 2, 14, 3, 4, 18, 19, 5)".
In captions for Figures in Supplementary Materials all abbreviations should be explained, as those figures should be self-explanatory.
The references are relatively old (only 21 of them have been published in last 10 years) and they are not presented in uniform style - in some references the full names of Journals are used and in others the abbreviations of the names of Journals are used. Moreover the reference 5 is lacking in references list and in references 9, 12, 18, 20 and 32 some bibliographic data are lacking - it should be corrected.
Author Response
Reviewer 1: Comments and Suggestions for Authors
The paper is interesting from both experimental and practical points of view and it could be potentially published in Biomedicines, but it needs some minor revisions before final acceptance.
Comment 1: In Abstract in line 8 the statement "and 5 each from the hospital" is not clear and it should be explained or corrected, while inline 11 the statement "activity of compound 5" should be used instead of "activity of 5".
Ans: Thank you for the comment. We have altered the “5 each from hospital" statement to “10 non-duplicate clinical strains” in Line 31-32. Also corrected “activity of 5” statement to “activity of compound 5” in Line 34.
Comment 2: Chapter 3 Materials and Methods should be placed between chapter 1 Introduction and chapter 2 Results and Discussion.
Ans: Thank you for the suggestions we have already placed Materials and Methods as Chapter 2.
Comment 3: As table 1 should be self-explanatory all abbreviations should be explained.
Ans: Explanatory note for each symbol has been added to the Abbreviations section.
Comment 4: In table 2 in the title and in description of particular columns the statements compound 1, compound 2, compound 3, compound 4 and compound 5 should be used instead of 1-5.
Ans: Thank you for the suggestion to improve the clarity and we have changed each compound name as instructed.
Comment 5: In subchapter 3.2 Synthesis of BODIPY derivatives the statements "Preparation of compound 8 (and similarly 1, 10, 12, 2, 14, 3, 4, 18, 19, 5)" should be used instead of "Preparation of 8 (and similarly 1, 10, 12, 2, 14, 3, 4, 18, 19, 5)1, 10, 12, 2, 14, 3, 4, 18, 19, 5)".
Ans: We have added the word “compound” ahead of each compound number.
Comment 6: In captions for Figures in Supplementary Materials all abbreviations should be explained, as those figures should be self-explanatory.
Ans: Explanatory note for each symbol has been added to the Abbreviations section.
Comment 7: The references are relatively old (only 21 of them have been published in last 10 years) and they are not presented in uniform style - in some references the full names of Journals are used and in others the abbreviations of the names of Journals are used. Moreover, the reference 5 is lacking in references list and in references 9, 12, 18, 20 and 32 some bibliographic data are lacking - it should be corrected.
Ans: Thank you very much for highlighting the mistakes. We have corrected the whole reference list by adhering to journal criteria.
Reviewer 2 Report
The manuscript fits the scope of the journal. The authors report important information for potential alternatives to antibiotics for combatting MRSA infections. The manuscript needs only minor revision before its acceptance for publication.
Line 358, the oxygen quantum yields for the tested compounds 1-3 are 0.07-0.09 according to table 1
Line 378, the @ should be replaced by m; the PDT mM values in Table 2 are between 25 and 100 mM
References: please adhere to the journal formatting guidelines (abbreviated journal name; year in bold).
Author Response
Reviewer 2: The manuscript fits the scope of the journal. The authors report important information for potential alternatives to antibiotics for combatting MRSA infections. The manuscript needs only minor revision before its acceptance for publication.
Comment 1: Line 358, the oxygen quantum yields for the tested compounds 1-3 are 0.07-0.09 according to table 1.
Ans: Thank you very much for highlighting the mistakes in this section. It has been amended accordingly.
Comment 2: Line 378, the @ should be replaced by m; the PDT mM values in Table 2 are between 25 and 100 mM
Ans: Thank you very much for highlighting the mistakes in this section. Actually the concentration values are withing micromolar values and have been amended accordingly.
Comment 3: References: please adhere to the journal formatting guidelines (abbreviated journal name; year in bold).
Ans: Thank you very much for highlighting the mistakes in this section. We have revised the whole reference section by adhering to the journal criteria.